# Wellness Perception of South Korean Elementary School Students during the COVID-19 Endemic

**DOI:** 10.3390/healthcare12010069

**Published:** 2023-12-28

**Authors:** Yongsuk Seo, Eui-Jae Lee, Jin-Young Kim, Jung In Yoo, Hyun-su Youn

**Affiliations:** 1Exercise Physiology Laboratory, Kookmin University, Seoul 02707, Republic of Korea; yseokss@gmail.com; 2Department of Physical Education, Graduate School of Education, Sogang University, Seoul 04107, Republic of Korea; hoho6468@sogang.ac.kr; 3Department of Sports in Life, Jangan University, Hwaseong-si 18331, Republic of Korea; kimjy6543@jangan.ac.kr; 4Department of Sports Science, The University of Suwon, Hwaseong-si 18323, Republic of Korea; 5Department of Physical Education, College of Education, WonKwang University, Iksan-si 54538, Republic of Korea

**Keywords:** elementary student, health perception, endemic

## Abstract

This study aimed to analyze health management awareness among South Korean elementary school students in COVID-19 endemic areas. Methods. Using convenience sampling, 675 South Korean elementary school students (age 11–12 years old) were selected as participants in July 2023. Data for the study were collected via online and offline surveys between July and August 2023. The collected data were subjected to frequency, reliability, and multicollinearity analyses, independent sample *t*-tests, and importance-performance analysis (IPA). Results. The findings indicated the following: (1) There was no significant difference in health management performance between male and female children. (2) Children who had not experienced COVID-19 infection, had a higher level of “hygiene management” performance. (3) Among children who did not wear masks during physical activity, “mental health management” and “physical activity management” performance were higher, while “hygiene management” performance was lower. (4) The IPA matrix analysis revealed that, compared to the COVID-19 pandemic period, “physical activity management”, “dietary habit management”, and “sleep management” still required improvement, while “hygiene management” and “disease management” appeared to have decreased due to the relaxation of epidemic control efforts. Conclusion. As per the study’s findings, schools, local communities, and families should make efforts to develop and implement preventive and individualized health management programs that consider the individual characteristics of their children.

## 1. Introduction

In May 2023, the World Health Organization declared the COVID-19 pandemic to be an endemic [1]. This came three years and four months after the pandemic declaration on 11 March 2020, and marked a reduction in the level of alertness and social restrictions [2]. The recent pandemic had created a disruption across the world since late 2019, instilling fear among humans. More than 700 million infections were reported [3] (https://data.who.int/dashboards/covid19/cases?n=c (accessed on 15 January 2023)), surpassing the approximately 500 million infections caused by the Spanish flu, previously regarded as the most severe pandemic of the 20th century. Among these, 6,284,871 lives were lost by 2022 [4]. However, scientists and researchers are now one step closer to completely eradicating COVID-19 [5].

Wellness is a multifaceted concept that encompasses various dimensions of an individual’s overall health and well-being, including physical, intellectual, emotional, social, spiritual, vocational, financial, and environmental aspects [6]. It is a dynamic, ever-changing process that involves the interplay of physical, mental, social, and environmental factors, aiming for a state of health and happiness rather than merely the absence of illness [7].

In May 2022, the South Korean government initiated a policy to lift mandatory precautionary measures such as the requirement to wear masks at gatherings of fewer than 50 people in outdoor settings. Furthermore, policies regarding mask-wearing and social distancing, both indoors and outdoors, have now been left to citizens’ discretion. However, healthcare facilities, with the goal of maintaining effective infection control on the frontlines, have not granted exemptions from mask-wearing [8].

This measure was welcomed by children and adolescents who spend most of their daily time at school, as it signified a step toward the recovery of educational activities that had been compromised during the COVID-19 pandemic. During the pandemic, the learning abilities, health, and social skills of children and adolescents progressively deteriorated [9]. A previous study on the health of children and adolescents during the COVID-19 pandemic suggested that mental health worsened during this period. Interestingly, the incidence of depression, anxiety, social isolation, maladjustment, stress, and deteriorating physical health has increased among children and adolescents [9]. Studies found that the prevalence of obesity has increased while physical activity among children and adolescents has significantly decreased, resulting in a decline in health-related quality of life (HRQoL) [10,11,12].

In this context, Lee, So, and Youn [13] conducted an importance-performance analysis (IPA) on the health perceptions of South Korean adolescents during the COVID-19 pandemic and reported an imbalance between therapeutic and preventive health factors [13]. The IPA method has found widespread application across diverse fields due to its simplicity and utility in structuring survey findings and devising effective strategies [14,15]. Subsequently, an analysis of health perception differences based on virtual physical education class types in the school setting, which transitioned to online learning, revealed unfavorable situations in terms of physical activity, dietary habits, and sleep management [16].

Studies conducted during the pandemic continued to show negative indicators of physical activity, dietary habits, and sleep management among adolescents [17]. On 2 May 2022, the Korean government’s partially relaxed containment policies were introduced, and a study was conducted with elementary school students; however, preventive health factors such as physical activity, dietary habits, and sleep management still showed low indicators. Furthermore, therapeutic health factors related to disease management and hygiene have tended to decrease in practice because of the impact of relaxed containment measures [8].

Therefore, this study analyzes the health care awareness of Korean children during the endemic period. Further, we also analyzed the importance and performance of health management among elementary school students based on their COVID-19 infection status, mask-wearing during physical activity, and sex differences in alignment with the endemic phase.

## 2. Material and Methods

### 2.1. Participants

The study was approved by the Wonkwang University IRB (WKIRB-202307-SB-048). A total of 675 South Korean elementary school students were selected as research participants for the study. The study participants were selected from four schools in the metropolitan area of South Korea; they were 11 to 12 years old, 5th and 6th grade students. Convenience sampling, a non-probability sampling technique, was employed, and the selected research participants were surveyed in offline and online formats using Google Forms in July 2023. For the online survey, Google Survey was employed, and for the offline survey, the researcher printed the survey and delivered it to elementary school teachers. The survey was administered during free time and after-school hours under the supervision of the classroom teacher. A total of 683 participants voluntarily took part in this study, but after excluding those who responded uniformly and those whose responses could not be verified, the final study sample comprised 675 individuals.

### 2.2. Instruments

In the study, we selectively utilized scales that aligned with the research objectives based on an analysis of prior research [18,19,20,21]. The general characteristics of the participants, such as sex, COVID infection status, and mask-wearing during physical activity, were constructed as nominal scales. The COVID-19 virus infection of all Korean citizens is confirmed through antibody testing. Therefore, the study participants’ history of COVID-19 virus infection is an antibody response and a past infection because at the time of conducting the survey, students infected with the coronavirus were not attending school. To measure health perception, we adapted and modified the scale based on Ware’s [18] health perception scale as validated for validity and reliability [19,20,21]. Specifically, health management was subdivided into mental health, disease, physical activity, sleep, dietary habits, and hygiene. Subsequently, a modified IPA technique was employed to empirically analyze the data [22,23].

### 2.3. Reliability of Instruments

To validate the scales used in this study, we employed Cronbach’s α to assess the internal consistency of the items. The results are summarized in Table 1. As in previous studies, six factors (with factor loadings above 0.5) emerged, and they were named as follows: dietary management, disease management, mental health, physical activity, sleep management, and hygiene management. The reliability analysis (Cronbach’s α) yielded values ranging from a minimum of 0.703 to a maximum of 0.927, demonstrating satisfactory levels of reliability [24].

### 2.4. Statistical Analysis

The collected data were analyzed using the Statistical Package for Social Science software (SPSS ^®^ version 19.0, IBM, Somers, NY, USA). The specific analytical procedure was as follows. First, multicollinearity was confirmed using a modified IPA analysis [25]. Second, independent sample *t*-tests were conducted to examine performance differences based on sex, COVID-19 infection status, and mask-wearing during physical activity. Third, a modified IPA method was used to validate the importance and performance of each variable.

## 3. Results

The demographic characteristics of the participants are presented in Table 2.

### 3.1. Health Management Performance by Sex

An independent sample *t*-test was conducted to examine sex-based differences in health management performance among South Korean elementary school students. The analysis indicated no significant differences in any of these aspects (Table 3).

### 3.2. Health Management Performance by COVID-19 Infection Status

An independent sample *t*-test was conducted to examine the differences in health management performance among South Korean elementary school students based on their COVID-19 infection status. Table 4 presents the results of the study. The analysis revealed statistically significant differences in the “hygiene management” aspect (*p* ≤ 0.01). It appears that students with no experience of infection were more diligently practicing “hygiene management” than those with the experience.

### 3.3. Health Management Performance by Mask-Wearing during Physical Activity

To investigate differences in health management performance among South Korean elementary school students based on mask-wearing during physical activity, a paired sample *t*-test was conducted. Table 5 presents the results. The analysis indicated statistically significant differences in “mental health”, “physical activity”, and “hygiene management”. In “mental health” and “physical activity”, the “non-wearing” group showed higher scores, while “hygiene management” had higher scores in the “wearing” group.

### 3.4. Importance-Performance Analysis of Health Management

To visually represent South Korean elementary school students’ perceptions of health management using factor-based IPA matrices, we divided the space into quadrants using the intersection of the median importance values of 1.20 and median performance values of 4.09 as the reference point. The results are illustrated in Figure 1. Table 6 presents the results. Mental health falls into Quadrant I, which is characterized by high importance and high performance (indicating the need to maintain current good work). Disease management was placed in Quadrant II, reflecting high performance but low importance (suggesting the need to focus on efforts here). Dietary management, physical activity, and hygiene management were all categorized in Quadrant III, which signifies low importance and performance, making them low-priority areas. Finally, sleep management was placed in Quadrant IV, with high importance but low performance, implying a potential need for adjustment (possibly overkilling).

## 4. Discussion

In this study, we analyzed South Korean elementary school students’ perceptions of health management in the context of the COVID-19 endemic phase, considering sex differences, COVID-19 infection status, and mask-wearing during physical activity. Additionally, we used a modified IPA method to examine the importance of health management and the extent to which participants currently practice it.

First, during the COVID-19 endemic phase, no significant difference was noted in the level of health management execution between sexes. South Korean education, based on gender equality, generally shows no significant differences in how boys and girls perceive and execute various aspects of health management [26]. This non-differentiation in gender perceptions and actions was being maintained even during the COVID-19 pandemic when online and offline classes coexisted [15].

However, the South Korean elementary school students demonstrated differences in health management based on their COVID-19 status during the endemic phase. It was observed that children who had not been infected with the virus were more diligent in their “hygiene management”. Generally, individuals who have been infected with COVID-19 and have recovered possess antibodies that provide immunity for a few months, reducing the likelihood of a reinfection [27]. In the past three years of the pandemic, many citizens, including children, were informed about COVID-19 and health [28]. The South Korean government, local authorities, schools, and other institutions have played essential roles in raising citizens’ health literacy through pandemic-related policies and education [29]. Therefore, during the COVID-19 pandemic, elementary school students who were not infected with SARS-CoV-2 were expected to continue to maintain hygiene management such as wearing masks, disinfecting hands, and social distancing because they continuously received COVID-19 prevention education during the pandemic.

Moreover, South Korean elementary school students exhibited differences in health management based on whether they wore masks during physical activities. Children who did not wear masks during physical activity exhibited higher levels of “physical activity” and “mental health”. This phenomenon can be attributed to research indicating that wearing masks during exercise can reduce physical activity levels [30] and individuals may experience increased feelings of depression and anxiety [31,32]. Students may also feel more psychologically liberated and engage in more physical activities when not wearing masks. Contrastingly, those who wore masks tended to be more concerned with hygiene management. Research on adults has shown that individuals with better subjective health tend to adhere to mask-wearing practices [33]. This suggests that children who diligently wear masks may also exhibit behaviors associated with health promotion, such as hygiene management.

Finally, students’ perceptions of the importance of health management and its actual implementation, as assessed using the IPA matrix, were compared between the COVID-19 pandemic and current endemic phases. First, “mental health management” was positioned in Quadrant I (Keep Up the Good Work) on the matrix. This quadrant represents factors for which both importance and execution are high, indicating that the area performs well and should be maintained [34]. During the initial COVID-19 pandemic period, when schools were closed and online learning was prevalent, “mental health management” was identified as an area requiring urgent improvement [13]. However, during the phase of relaxed pandemic measures and a combination of online and offline classes, research conducted on children found that “mental health management” was categorized as an area where both, importance and execution were high, suggesting that it should be maintained [16]. This suggests that fear of the virus, compliance with preventive measures, and social isolation during the pandemic gave way to a sense of mental relief and autonomy, particularly in the management of mental health. Consequently, with mask-wearing and other preventive measures now being voluntarily relaxed and schools transitioning to full-time offline classes, the current endemic phase now allows children to place greater importance on mental health management and improve their practices. Second, “disease management” was identified in Quadrant II (Concentrate Here) on the matrix. This quadrant represents factors where the perception of importance is low but the level of execution is high, indicating that this area requires urgent improvement in the perception of importance [34]. In the early stages of the COVID-19 pandemic, “disease management” was universally high for adults, adolescents, and children, irrespective of age, and the level of implementation was substantial [13]. Even though the pandemic has now reached a state of lull, adolescents and children continue to exhibit a strong commitment to disease management [16,35]. However, with the lower mortality rate of the omicron variant of coronavirus and the relaxation of public health measures, the importance of disease management in children appears to have diminished. Nevertheless, individuals who recognized the significance of disease management during the pandemic perceive it as highly meaningful. In Quadrant III (Low Priority), “physical activity management”, “hygiene management”, and “diet management” were identified. These factors have low importance perception and execution levels among the students, indicating the need for long-term efforts to improve them [34]. “Physical activity management” and “diet management” were factors that children and adolescents generally did not consider important and did not execute, regardless of the severity or relaxation phase of the COVID-19 pandemic [13,15,16,36]. This can be attributed to a sedentary lifestyle and relying on food deliveries that persisted during the past three years of the pandemic [37,38]. Physical inactivity affects not only children’s physical growth, but also their social relationships and brain development [37]. Recent reports have suggested that infection with the COVID-19 virus can lead to serotonin depletion, often referred to as the “happy hormone” [39,40]. Most children infected with the COVID-19 virus have experienced severe limitations in daily function with fatigue, dyspnea, and concentration difficulties [41].

The multifaceted results of children’s health management obtained in this process can serve as essential foundational data for planning school education and community and family initiatives, as well as for the development and implementation of policies and programs by educational authorities.

Therefore, physical activity must be actively promoted to restore and activate serotonin levels [40,41]. In this context, the Ministry of Education in South Korea is pursuing various policies, including physical education, to restore and activate children’s physical activity. This effort was viewed as a positive educational direction. In Quadrant IV (Possible Overkill), “sleep management” was explored. This quadrant signifies factors where the perception of importance is high but the level of execution is low, indicating an urgent need for improvement in the execution level [34]. During the pandemic, “sleep management” was one of the factors that required improvement among children and adolescents, along with “physical activity” and “diet” [13,16]. Excessive exposure to digital media due to online classes was the primary reason children and adolescents did not effectively manage their sleep well [42,43]. With the current transition to full-time in-person classes, children still recognize the importance of “sleep management” due to the habits they developed during the pandemic, but the performance remains inadequate.

The current study has certain limitations. First, this study focused solely on elementary school student during the early phase of the endemic, which may not be representative of other age groups and regions. Additionally, the study would have been more significant if it had been conducted longitudinally or involved a comparison between the period of the pandemic and the post-pandemic phase.

In future research, it is essential to conduct studies related to health management during the COVID-19 endemic period for different age groups, including infants, adolescents, adults, and older adults. Additionally, it is necessary to expand the scope of this research by comparing the results of studies from different countries and regions. From a research methodology perspective, conducting qualitative research methods, such as in-depth interviews, participant observations, or mixed research methods, can provide more comprehensive and persuasive research results and insights.

## 5. Conclusions

This study aimed to analyze the importance and performance of health management among South Korean children during the COVID-19 pandemic. The results indicated the following: As differences in the level of health management were observed based on the infection status and mask-wearing during activities, efforts should be made to develop and implement personalized health management programs in schools, local communities, and households, considering the individual characteristics of children. Furthermore, it was noted that preventive health management for children centered on physical activity, dietary habits, and sleep management, remained low even during the COVID-19 pandemic. Therefore, continuous monitoring, program development, and implementation efforts are necessary in schools, communities, and households.

## Figures and Tables

**Figure 1 healthcare-12-00069-f001:**
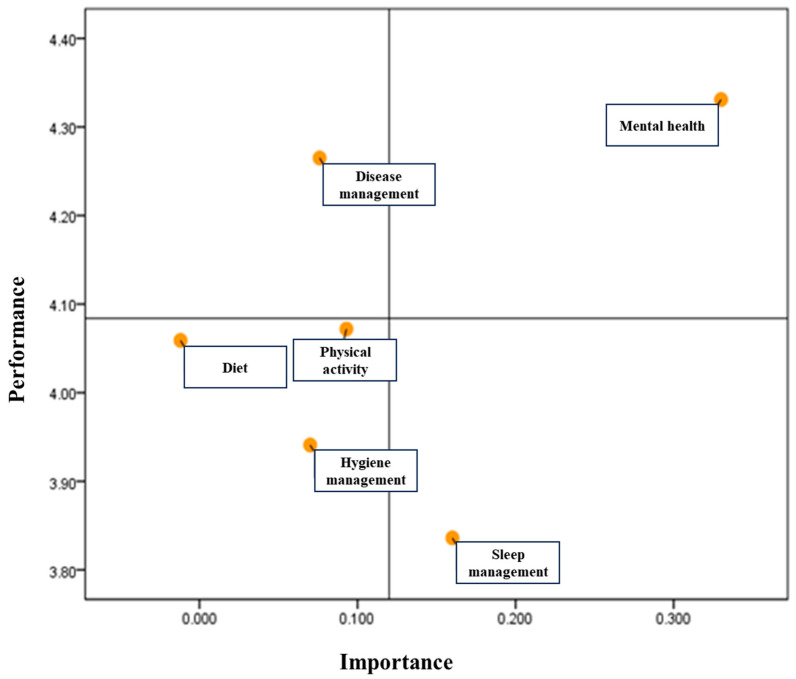
IPA matrix—factors.

**Table 1 healthcare-12-00069-t001:** Inter-item consistency.

	Dietary	Disease	Mental Health	Physical Activity	Sleep	Hygiene
Dietary management 3 Adequate quantity of food	0.915 *	0.080	0.130	0.123	0.111	0.090
Dietary management 2 Dietary habits	0.889 *	0.154	0.126	0.108	0.128	0.112
Dietary management 1 Regular meal	0.883 *	0.136	0.119	0.112	0.172	0.050
Disease management 3 Preventive care	0.068	0.830 *	0.054	0.139	0.064	0.102
Disease management 2 Prescription control	0.155	0.815 *	0.174	0.037	0.149	0.130
Disease management 1 Disease prevention	0.141	0.723 *	0.179	0.081	0.234	0.144
Mental health 2 Trust	0.105	0.135	0.792 *	0.141	0.065	0.032
Mental health 1 Happiness	0.106	0.089	0.763 *	0.071	0.197	0.048
Mental health 3 Sense of belonging	0.137	0.152	0.692 *	0.224	0.116	0.136
Physical activity 1 Sports activity	0.071	0.051	0.236	0.782 *	0.034	0.016
Physical activity 2 Regular physical activity	0.115	0.054	0.232	0.781 *	0.105	0.000
Physical activity 3 Physical movement	0.116	0.133	−0.032	0.760 *	0.115	0.075
Sleep management 2 Sleep hygiene	0.085	0.129	0.136	0.066	0.768 *	−0.015
Sleep management 1 Regular sleep	0.116	0.138	0.057	0.157	0.766 *	0.095
Sleep management 3 Sleep environment	0.157	0.105	0.155	0.034	0.722 *	0.097
Hygiene management 1 Wearing mask	0.056	0.016	0.014	−0.017	0.024	0.872 *
Hygiene management 2 Social distancing	0.068	0.132	−0.003	0.056	0.182	0.784 *
Hygiene management 3 Sanitary practice	0.098	0.215	0.212	0.062	−0.029	0.650 *
Eigen value	2.592	2.106	2.001	1.979	1.954	1.916
Explained (%)	14.401	11.703	11.119	10.993	10.857	10.645
Cumulative (%)	14.401	26.103	37.222	48.215	59.072	69.717
Cronbach’s α	0.927	0.789	0.725	0.719	0.704	0.703

Note: * Cronbach’s α is satisfactory levels of reliability.

**Table 2 healthcare-12-00069-t002:** Participants’ demographic characteristics.

Classification			Cases (n)	Percentage (%)
Sex	Boys		340	50.4
	Girls		335	49.6
Age (years)	Boys	11 years	160	47.0
		12 years	180	53.0
	Girls	11 years	165	49.2
		12 years	170	50.8
COVID-19 infection status	Yes		590	87.4
	No		85	12.6
Wearing masks during physical activity	Wearing		294	43.6
	Not Wearing		381	56.4
Total			675	100

**Table 3 healthcare-12-00069-t003:** Analysis of health management performance by sex.

	Sex	N	Mean	St Dev	t	*p*
Mental health	Boys	340	4.334	0.698	0.131	0.896
	Girls	335	4.327	0.681		
Disease management	Boys	340	4.208	0.865	−1.862	0.063
	Girls	335	4.323	0.744		
Physical activity	Boys	340	4.103	0.891	0.933	0.351
	Girls	335	4.041	0.838		
Sleep management	Boys	340	3.843	0.968	0.199	0.843
	Girls	335	3.829	0.899		
Dietary management	Boys	340	4.002	0.992	−1.532	0.126
	Girls	335	4.117	0.966		
Hygiene management	Boys	340	3.902	0.959	−1.124	0.261
	Girls	335	3.981	0.867		

**Table 4 healthcare-12-00069-t004:** Analysis of health management performance by COVID-19 infection status.

	Infection Status	N	Mean	St Dev	t	*p*
Mental health	Yes	590	4.344	0.684	1.312	0.190
	No	85	4.239	0.723		
Disease management	Yes	590	4.281	0.801	1.322	0.187
	No	85	4.157	0.857		
Physical activity	Yes	590	4.075	0.864	0.240	0.810
	No	85	4.051	0.883		
Sleep management	Yes	590	3.828	0.936	−0.572	0.568
	No	85	3.890	0.922		
Dietary management	Yes	590	4.050	0.975	−0.627	0.531
	No	85	4.122	1.018		
Hygiene management	Yes	590	3.906	0.920	−2.632	0.009 *
	No	85	4.184	0.841		

* *p* ≤ 0.01.

**Table 5 healthcare-12-00069-t005:** Analysis of health management performance by mask-wearing during physical activity.

Measured Variable	Infection Status	N	Mean	St Dev	t	*p*
Mental health	Wearing	294	4.2528	0.724	−2.595	0.010 *
	Not Wearing	381	4.3911	0.655		
Disease management	Wearing	294	4.2290	0.786	−1.021	0.308
	Not Wearing	381	4.2931	0.825		
Physical activity	Wearing	294	3.9705	0.868	−2.691	0.007 *
	Not Wearing	381	4.1505	0.856		
Sleep management	Wearing	294	3.7823	0.946	−1.315	0.189
	Not Wearing	381	3.8775	0.922		
Dietary management	Wearing	294	4.0658	0.967	0.151	0.880
	Not Wearing	381	4.0542	0.991		
Hygiene management	Wearing	294	4.2812	0.736	9.263	0.000 *
	Not Wearing	381	3.6789	0.953		

* *p* ≤ 0.05.

**Table 6 healthcare-12-00069-t006:** Distribution of health perception factors.

Quadrant	Criteria	Variable Distribution
Quadrant I	Importance ↑ Performance ↑	Mental Health management(Happiness, Sense of belonging, Trust)
Quadrant II	Importance ↓ Performance ↑	Disease management(Disease prevention, Prescription control, Preventive care)
Quadrant III	Importance ↓ Performance ↓	Physical activity(Sports activity, Physical movement, Regular physical activity) Diet(Regular meal, Dietary habits, Adequate quantity of food) Hygiene management(Wearing mask, Social distancing, Sanitary practice)
Quadrant IV	Importance ↑ Performance ↓	Sleep management(Regular sleep, Sleep hygiene, Sleep environment)

## Data Availability

The data presented in this study are available upon request from the corresponding author. The data are not publicly available owing to the protection of personal information.

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
