# Peer review of "Wellness Perception of South Korean Elementary School Students during the COVID-19 Endemic"

_healthcare, 2023, doi:10.3390/healthcare12010069_

Round 1

Reviewer 1 Report

Comments and Suggestions for Authors

The authors (AA) aim to analyse health management awareness among South Korean elementary school students in endemic areas.

This article could be useful to increase our knowledge of the issue. Addressing the issues below could make this manuscript eligible for publication.

Throughout the manuscript AA should be careful to refer to SARS-CoV-2 virus and SARS-CoV-2 infection, COVID-19 is the disease. Check it.

Abstract

AA should indicate the age range of students eligible for study.

What is the meaning of convenience sampling?

Indicate when the online and offline surveys were carried out.

Introduction

By WHO pandemic declaration was on 11 March 2020 (https://www.who.int/director-general/speeches/detail/who-director-general-s-opening-remarks-at-the-media-briefing-on-covid-19---11-march-2020), not January 2020. Modify it.

It has caused over 527,971,809 infections, surpassing 500 million infections caused by the Spanish flu, which was considered the worst pandemic of the 20th century.” I suggest indicating an approximate number of infections given the speed at which they are increasing and AA should indicate the source.

However, the scientists and researchers are now one step closer to completely eradicating COVID-19” Add reference.

This step has been welcomed by children and adolescents who spend most of their daily time in school, as it signifies a step toward the recovery of educational activities that had been compromised by the COVID-19 pandemic. During the pandemic, the learning abilities, health, and social skills of children and adolescents have progressively deterio[1]rated [3]. A previous study on the health of children and adolescents during the COVID[1]19 pandemic suggested that mental health worsened during this period. Interestingly, the incidence of depression, anxiety, social isolation, maladjustment, stress, and deteriorating physical health has increased among children and adolescents [3]. Studies have found that the prevalence of obesity has increased while physical activity among children and adolescents has significantly decreased, resulting in a decline in Health-Related Quality of Life (HRQoL) [4,5]” AA could add references such as:

Saulle R, De Sario M, Bena A, Capra P, Culasso M, Davoli M, De Lorenzo A, Lattke LS, Marra M, Mitrova Z, Paduano S, Rabaglietti E, Sartini M, Minozzi S. School closures and mental health, wellbeing and health behaviours among children and adolescents during the second COVID-19 wave: a systematic review of the literature. Epidemiol Prev. 2022 Sep-Dec;46(5-6):333-352. doi: 10.19191/EP22.5-6.A542.089.

Partially relaxed containment policies were introduced, and a study was conducted with elementary school students; however, preventive health factors like physical activity, dietary habits, and sleep management still showed low indicators.” Add reference. Partially relaxed containment policies were introduced… when?

“… this study aimed to analyse the health management perceptions of Korean children and adolescents from the early stages of the COVID-19 pandemic to the mitigation phase” Why do the authors talk about adolescents if the study is carried out in elementary schools? If the survey was carried out only once in June 2023, how can we understand the trend from the early stages of the COVID-19 pandemic to the mitigation phase?

The multifaceted results of children’s health management obtained in this process can serve as essential foundational data for planning school education and community and family initiatives, as well as for the development and implementation of policies and programs by educational authorities” Move this part into the discussion.

AA should better explain their aim.

Materials and methods

Which and how many elementary schools are involved in the study?

Explain convenience sampling. What are inclusion and exclusion criteria?

Explain better how the survey is administered. (online/offline)

Specify how the infection status is considered. via clinical diagnosis or antibody response or other? is it a previous or ongoing infection or both?

Throughout the manuscript change gender with sex.

Add analysis by age.

Table 1: Move into results. Add age.

“In this study, we selectively utilized scales that aligned with the research objectives based on an analysis of prior research.” Add reference.

Specify modified IPA.

What do the authors mean by health management?

AA should report details of the survey. What are the questions? Were the questions appropriate for children of all primary school age groups? or were 5/6 year olds not included? or was it compiled with the support of parents or teachers?

Results

Table 3/4/5. Specify the acronym M and SD.

“… more diligently practicing “hygiene management” than those with an experience.” More diligently… what does it mean? What criteria are included? Number of times you sanitize your hands, use of daily gel, use of gloves?

Discussion

Therefore, even during the COVID-19 endemic phase, it is assumed that elementary school students, who have not been infected with the SARS-CoV-2 continue to practice measures such as wearing masks, hand hygiene, and social distancing based on their learned health information.” Explain better this.

This suggests that children who diligently wear masks may also exhibit behaviours associated with health promotion, such as hygiene management.” The study cited reports data about an adult population, how does it relate to children?

“Finally, the students’ perceptions of the importance of health management and its actual implementation, as assessed using the IPA matrix, were compared between the COVID-19 pandemic and current endemic phases. First, “mental health management” was positioned in Quadrant I (Keep Up the Good Work) on the matrix. This quadrant represents factors for which both importance and execution are high, indicating that the area performs well and should be maintained [24]. During the initial COVID-19 pandemic ……….” This part is unclear. It is not explained whether their study is the continuation of a previous study and for this reason they write about a trend. Clarify the entire paragraph.

Author Response

Thank you for your good opinion. I revised and supplemented it.

Reviewer 2 Report

Comments and Suggestions for Authors

Thank you for the opportunity to review the article "Health Management Awareness of South Korean Elementary School Students during Endemic".
Health management of children and adolescents is among the priorities in health policy. It has become especially important during the pandemic period.
The research paper raises many questions and lacks clarification on several issues:
1.In what period the study was conducted and on what basis the study participants were selected;
2.The research section lacks information on what factors went into the 6 main research areas;
3.The research areas only referred to the comparison in wearing or not wearing masks;
4. The summary results in Table 2 do not add anything relevant.
5. The matrix data should be made more specific. The authors need to explain more clearly what data elements are of interest in each subsection and what the figures represent in each case.

Author Response

(The authors gave the same response as above.)

Reviewer 3 Report

Comments and Suggestions for Authors

This article mainly analyzes the health management awareness of Korean children and adolescents in the early to remission stages of the COVID-19 pandemic and takes into account gender differences, COVID-19 infection status, and the wearing of masks during physical activity.

The authors of this article focus on the highly variable time period from the early to the remission phase of the COVID-19 pandemic and explore the trends in health management perceptions of Korean children and adolescents during this time period. In addition, the authors have adapted and improved Ware's Perception of Health Scale by subdividing health management into six blocks: mental health, illness, physical activity, sleep, eating habits, and hygiene, and used a modified IPA technique to examine the importance of health management and the extent of current practice, which is highly innovative and scientifically sound. Another highlight of this paper is that at the end of the discussion section of the article, the authors also present their ideas about following up with further research and refinement, giving some direction for future research in this area.

However the article also has the following points that could be improved:

1. The main endemic in this article refers to the COVID-19 pandemic , this could have been made clearer in the title and abstract section.

2. The sampling method of this article adopts convenience sampling, whether the author can elaborate on why this sampling method is adopted and the errors that may be brought about by using this method in the research method or discussion section.

3. In Table 3 and Table 4, the "M" representing the Mean should be written as "Mean" in detail, so that readers can have a clearer understanding.

4. A discussion of the limitations of this article could be added to the discussion section.

5. In response to the finding that "the perceived importance and level of implementation of the three factors of physical activity management, hygiene management and dietary management are low among students", the authors could give some more specific measures and recommendations to address the issue.

Comments on the Quality of English Language

Minor editing of English language required.

Author Response

(The authors gave the same response as above.)

Reviewer 4 Report

Comments and Suggestions for Authors

Thank you for sending me this paper to review.  

The study title includes the term 'health management awareness'.  This term should either be defined or you could consider changing it to something more recognisable like 'well being'.  

Figure 1: I was unclear about 'sanitary management'.  It did not seem to have been investigated in the data.  Does it refer to 'hygiene' data? 

Can you explain how the sample was recruited?  You say 675 students were selected to participate and 675 agreed.  This 100% response rate is very unusual.  Was their participation voluntary? How were they 'selected'? Was there a cohort of students who were not 'selected' and why.  

Data were collected using a survey design.  Did the students know the data collectors, and if they did how did you manage social desirability bias?

This convenient sample consisted of students from an elementary school in South Korea which may not generate generalisable findings.  In terms of trying to make the results more generalisable, it may have been useful to compare characteristics of these students to characteristics of all elementary students in South Korea.  Perhaps using a national database of student characteristics.

You will see I ticked 'ethical concerns' and 'scientific soundness' as low.  The reason for this relates to how the data were collected, and how the sample was selected.  Providing a short explanations at the relevant sections ie extraordinarily high response rate, selection of sample, voluntariness, and social desirability in the paper will address these comments.  

Conducting a comparison of the sample characteristics with characteristics of a national sample of students may help the generalisability of findings.  

I hope you find these comments helpful.  

Author Response

(The authors gave the same response as above.)

Round 2

Reviewer 1 Report

Comments and Suggestions for Authors

The authors (AA) have addressed some reviewers' comments. Overall the changes made have improved the manuscript but there are still some points to improve.

Abstract

Point number (N)

N 1: Please indicate in the abstract the age range of students eligible for study.

N 3: Please indicate when the online and offline surveys were carried out instead of the month of participants’ selection, so please modify the sentence.

Introduction

N 5: Please add reference such as https://covid19.who.int/ and to date there are over 700 million cases.

N 9: Please add reference.

N 10: The aim seems a bit confusing; please answer to the following questions: why do the authors talk about adolescents if the study is carried out in elementary schools? If the survey was carried out only once in June 2023, how can we understand the trend from the early stages of the COVID-19 pandemic to the mitigation phase? Moreover, the answer given from AA to this points is different from the text, so please try to make aim more clear.

N 12: please explain better the aim.

Materials and methods

N 14: please insert in the text the explanation of convenience sampling. What are inclusion and exclusion criteria?

N 16: Please add in the text your answer to clarify whether both “antibody response” and “past infection” must be present at the same time or not; if not please modify the sentence. And what about vaccinated children? An antibody response may also be due to vaccination. Please clarify when and how children did antibody test for this research.

N 19: Please add age in table 1 or in the text. Table 1 should be moved in the results section.

N 21: Define acronym IPA.

Results

N 25: Please answer to these questions: “… more diligently practicing “hygiene management” than those with an experience.” More diligently… what does it mean? What criteria are included? Number of times you sanitize your hands, use of daily gel, use of gloves?

Discussion

N 26: Please add reference if possible.

N 28: The answer is unclear as well as presented in the introduction given the presented methods and results. It is not explained whether their study is the continuation of a previous study and for this reason, they write about a trend. Clarify this issue.

Author Response

Thank you for your good review. Our researchers made general revisions reflecting the reviewer's corrective instructions, marked with yellow highlights.

Reviewer 2 Report

Comments and Suggestions for Authors

In my opinion, the corrections made by the authors allow for a positive review. 

Author Response

(The authors gave the same response as above.)
